# Artificial intelligence applications used in the clinical response to COVID-19: A scoping review

**Sean Mann** [1]*, **Carl T. Berdahl**[1,2], **Lawrence Baker**[1], **Federico Girosi**[1]

**1** RAND Corporation, Santa Monica, California, United States of America, **2** Departments of Medicine and Emergency Medicine, Cedars-Sinai Medical Center, Los Angeles, California, United States of America

* smann@rand.org

**Data Availability Statement:** All data are in the manuscript and/or supporting information files.

**Funding:** This work was funded by the Patient-Centered Outcomes Research Institute (PCORI, https://www.pcori.org/) under Contract No. IDIQ-

## Abstract

Research into using artificial intelligence (AI) in health care is growing and several observers predicted that AI would play a key role in the clinical response to the COVID-19. Many AI models have been proposed though previous reviews have identified only a few applications used in clinical practice. In this study, we aim to (1) identify and characterize AI applications used in the clinical response to COVID-19; (2) examine the timing, location, and extent of their use; (3) examine how they relate to pre-pandemic applications and the U.S. regulatory approval process; and (4) characterize the evidence that is available to support their use. We searched academic and grey literature sources to identify 66 AI applications that performed a wide range of diagnostic, prognostic, and triage functions in the clinical response to COVID-19. Many were deployed early in the pandemic and most were used in the U.S., other high-income countries, or China. While some applications were used to care for hundreds of thousands of patients, others were used to an unknown or limited extent. We found studies supporting the use of 39 applications, though few of these were independent evaluations and we found no clinical trials evaluating any application's impact on patient health. Due to limited evidence, it is impossible to determine the extent to which the clinical use of AI in the pandemic response has benefited patients overall. Further research is needed, particularly independent evaluations on AI application performance and health impacts in real-world care settings.

## Author summary

In this study we describe the use of artificial intelligence (AI) in the clinical response to COVID-19. AI has been variously predicted to play a key role during the pandemic or has been reported to have had little or no impact on patient care. Our findings support a balanced view. We identified 66 applications—specific AI products or tools—used in a variety of ways to diagnose, guide treatment, or prioritize patients during the pandemic response. Many were deployed early in 2020 and most were used in the U.S., other high-income countries, or China. Some were used to care for hundreds of thousands of patients though most were adopted at smaller scales. We found evaluation studies that supported

TO#22-RAND-ENG-AOSEPP-04-01-2020. All statements, findings, and conclusions in this publication are solely those of the authors and do not necessarily represent the views of the Patient-Centered Outcomes Research Institute (PCORI). The funders advised on study design. The funders played no role in data collection and analysis, decision to publish, or preparation of the manuscript.

**Competing interests:** The authors have declared that no competing interests exist.

the use of 39 of these applications, though few of these evaluations were written by independent authors, not affiliated with application developers. We found no clinical trials that evaluated the effect of using an AI application on patient health outcomes. Future research is needed to better understand the impact of using AI in clinical care.

## Introduction

### Background

Research into artificial intelligence (AI) for health care has grown rapidly, accompanied by a substantial increase in the number of AI applications receiving U.S. Food and Drug Administration (FDA) clearance since 2016.[1,2]. However, adoption has been limited, even in the field of radiology [3], which has been a particular focus of AI development [4]. The COVID-19 pandemic has been a proving ground for other health technologies, with an over 30-fold increase in U.S. telehealth usage in the first year of the pandemic [5] and the global deployment of novel mRNA vaccines [6]. Despite predictions that AI could play a key role in the clinical response to COVID-19 [7–9], there is little information on the range and extent of AI deployment during the pandemic.

Previous studies on the role of AI in the clinical response to COVID-19 have reviewed possible use cases [10,11], challenges to deployment [12–14], potential impacts on health equity [15], and recommendations to improve AI model quality [16,17]. Two separate reviews of academic literature published early in the pandemic identified hundreds of new AI models proposed to address COVID-19, though they did not attempt to identify which models, if any, had been deployed in patient care. These reviews found that all proposed models were flawed due to methodology, potential bias, or poor reporting, and the authors recommended that none be used in clinical practice [16,18]. Nevertheless, some AI applications have been deployed in clinical care for COVID-19 [19–22]. Despite this, no published review has identified more than a handful of applications that have progressed beyond development and testing to be used in clinical practice.

### Objectives

This study seeks to comprehensively identify and characterize AI applications—that is, specific AI-based products or tools—used in the clinical response to COVID-19. To do this, we analyzed a wide range of sources to answer the following four questions, which have not been addressed by other reviews:

1.  What AI applications were used in the clinical response to COVID-19?

2.  When, where, and at what scale were these applications used?

3.  How do these applications relate to technology in use before the pandemic, and to what extent have applications adopted in the U.S. been reviewed by the FDA?

4.  What evidence is publicly available to support these applications' use in clinical care?

## Methods

We performed a scoping review of academic and grey literature to identify a comprehensive set of literature that would answer the four key questions listed above. According to published guidance, a scoping view approach is well suited to exploratory analysis of broad topics

concerning the "extent, range, and nature of research activity" in a field [23]. We have followed the reporting guidelines contained in the Preferred Reporting Items for Systematic reviews and Meta-Analyses extension for Scoping Reviews (PRISMA-Scr) [24].

Our approach consisted of four steps: (1) consulting stakeholders; (2) document search and screening; (3) reviewing documents to identify AI applications; and (4) extracting information on AI applications and evaluation studies.

The topic of this article—the use of AI in the clinical response to COVID-19—was established by the sponsor prior to the start of the study as part of a broader project examining AI, COVID-19, and health equity. Stakeholder consultation, initial document searches, and document screening were undertaken as part of this broader project.

For the purposes of this study, we considered an application to be clinical in nature if it was used in efforts to improve patient health at the individual level as part of patient evaluation, clinical decision-making, or treatment delivery.

## Consulting stakeholders

At the outset of the study process, we engaged with a diverse set of health care stakeholders: a patient advocate, two clinicians, one health system representative, one insurer representative, one public policymaker, one public health official, one industry representative, and one researcher. We conducted separate semi-structured interviews with each of these stakeholders to elicit their recommendations on key questions, study design, and documents to review. We also asked stakeholders for information on any AI applications used in the COVID-19 response. Stakeholder inputs provided a preliminary set of documents and applications to consider for inclusion in our review.

Stakeholder consultation interviews were determined to be exempt by the RAND Corporation Human Subjects Protection Committee (HSPC ID# 2021-N0625), which serves as RAND's IRB. We obtained informed consent from stakeholder participants orally at the outset of all interviews. The interview protocol, which was provided to stakeholders and also covered topics related to AI and health equity as part of a broader research study, is available in S1 Appendix.

## Document search and screening

We searched several databases in December 2021 to identify documents describing the use of AI in the COVID-19 response. The search was limited to material that became available on or after January 1, 2020, the day after China first reported the possibility of a new virus to the World Health Organization [25].

We used structured search terms under the guidance of a medical reference librarian to identify academic review articles in PubMed, Web of Science, and IEEE Xplore Digital Library. We then adapted search terms to facilitate searches for gray literature documents, including news articles, clinical trials, and government documents on Proquest US Newsstream, Academic Search Complete, ClinicalTrials.gov, and the U.S. Food and Drug Administration (FDA) document library. We searched for documents that contained both an AI- and a COVID-19-related term. Additional searches were conducted to obtain documents relating to AI and health equity; while these searches were designed for use in a broader research effort, all results were screened for relevance to this study as well. Details on all searches are provided in supporting information file S1 Appendix.

We screened academic articles based on title and abstract as well as clinical trial records based on summary, description, and intervention fields. English-language documents that

discussed AI applications used in the response to COVID-19 were screened as eligible for full text review. We also screened in documents on AI and equity as part of the broader research effort.

To ensure consistency in screening decisions, we used dual-review methods and assessed inter-reviewer reliability. This began with three members of the project team each independently examining a random sample of approximately 10% of search results, followed by discussion of discrepancies and refinement of screening criteria. We then used these finalized criteria to conduct single-screening of the remaining search results, with random dual-review checks of 25% of the remaining documents to ensure continued consistency in the screening process. Disagreements in the two reviewers' screening decisions were resolved by a third project team member, with discussion of edge cases conducted on an as-needed basis.

## Reviewing documents to identify AI applications and evaluation studies

A member of the study team read the full text of each document that passed initial screening to identify AI applications used in the clinical response to COVID-19 response. Additional documents were obtained by following chains of relevant citations and hyperlinks from the initial set of documents as well as in the first 5 pages of results from two google searches: 'FDA-approved artificial intelligence COVID' and 'artificial intelligence clinical adoption COVID'.

Following identification of an AI application that was potentially used in the clinical response to COVID-19, we conducted targeted google searches using the application name and/or developer to obtain additional information and confirm its use.

To be included in our review, an application had to meet the following three criteria:

1. The application performed a clinical function, meaning that it aimed to improve patient health at the individual level as part of patient evaluation, clinical decision-making, or treatment delivery. Applications used to assist biomedical research, drug development, and public health policymaking were not included.

2. The application was used in the clinical response to COVID-19, whether to aid in prevention, diagnosis, prognosis, or treatment of the disease. Any documented use in patient care was sufficient for inclusion, even if only for a limited period of time or in limited settings. Applications that were made directly available to patients using broad distribution channels (e.g. government health websites, symptom checker platforms with a large user-base, medical testing sites) were included even if we did not find documentation on the extent of application use. Documented plans, recommendations, or government approvals for application deployment were not sufficient for inclusion.

3. The application used AI. We included all applications whose developers described them as using AI, "machine learning", or "deep learning". We also included all applications that reported use of specific machine learning (ML) methods, including advanced ML (e.g. neural networks, support vector machines, and complex decision trees), traditional ML (e.g. logistic or linear regression prediction models), and unsupervised ML (e.g. clustering, principal components analysis).

For each application we also examined company websites and conducted targeted searches of Google Scholar, examining the first 30 results for each search, to obtain publicly available evaluation studies. We searched both peer-reviewed articles as well as a wide variety of grey literature documents, including academic pre-prints, conference abstracts, company web pages, and regulatory documents, to identify evaluation studies on applications. We only counted a

document as an evaluation study if it reported information on evaluation results, study population, and outcome measurement methods.

### Extracting information on AI applications and evaluation studies

We extracted information on AI applications using a 65-item data collection form. All documents found on an application were reviewed during this data extraction process.

While prior studies have categorized AI applications developed to address COVID-19 in a variety of ways [11,16,18,21], no consensus framework exists. We developed our framework by identifying categories of applications that shared a common function in an inductive and iterative process. We proceeded in order of category size: we identified a large group of similar applications, assigned them a category label, and then considered the next largest group of applications. Applications that could belong to multiple groups were assigned to the largest category. We identified five functional categories of applications in this way and assigned the remaining applications to a catch-all 'Other' category.

We determined an application's extent of use based on the number of patients which an application was reportedly used to care for in the clinical response to COVID-19. We also counted other measures of reported usage, if these were likely to be within the same order of magnitude as the number of patients cared for. Number of COVID-19 cases or number of CT scans analyzed, for example, were counted, though we did not count the number of data points or CT scan slices analyzed when those were the only measure reported. We categorized the location of application use according to the Organization for Economic Cooperation and Development's high-income country (HIC) and low or middle-income country (LMICs) categories [26].

We extracted information on evaluation studies using an 14-item data collection tool. A primary reviewer extracted information on each study and a secondary reviewer then reviewed the abstraction. Disagreements were rare and were resolved in discussion between the two reviewers. We provide information extracted on each study together with verbatim text excerpts that served as the basis for extraction in supporting information file S2 Table.

## Results

Our searches of academic and gray literature databases yielded a total of 1,880 unique documents. After applying inclusion criteria at the level of the title and abstract, we screened 634 documents of potential interest to include in our review. The results of the document search and screening process are shown in Fig 1.

We identified 66 AI applications used in the COVID-19 response and grouped them into 6 functional categories. Information on individual applications is provided in the supporting information file S1 Table. A summary is provided in Table 1.

### Data inputs, context of use, and proposed benefits by application category

Applications within the same functional category tended to share similar data inputs, predicted variables, users, settings, and proposed benefits, as discussed below.

### Lung evaluation

Lung evaluation applications assessed X-ray images (n = 7), CT images (n = 12), or both (n = 1) to evaluate the lungs of patients with suspected or confirmed COVID-19. These applications assessed one or more of the following: presence of pneumonia (n = 10), pneumothorax (n = 2), other lung abnormalities associated with COVID-19 (n = 12), or an overall COVID-19

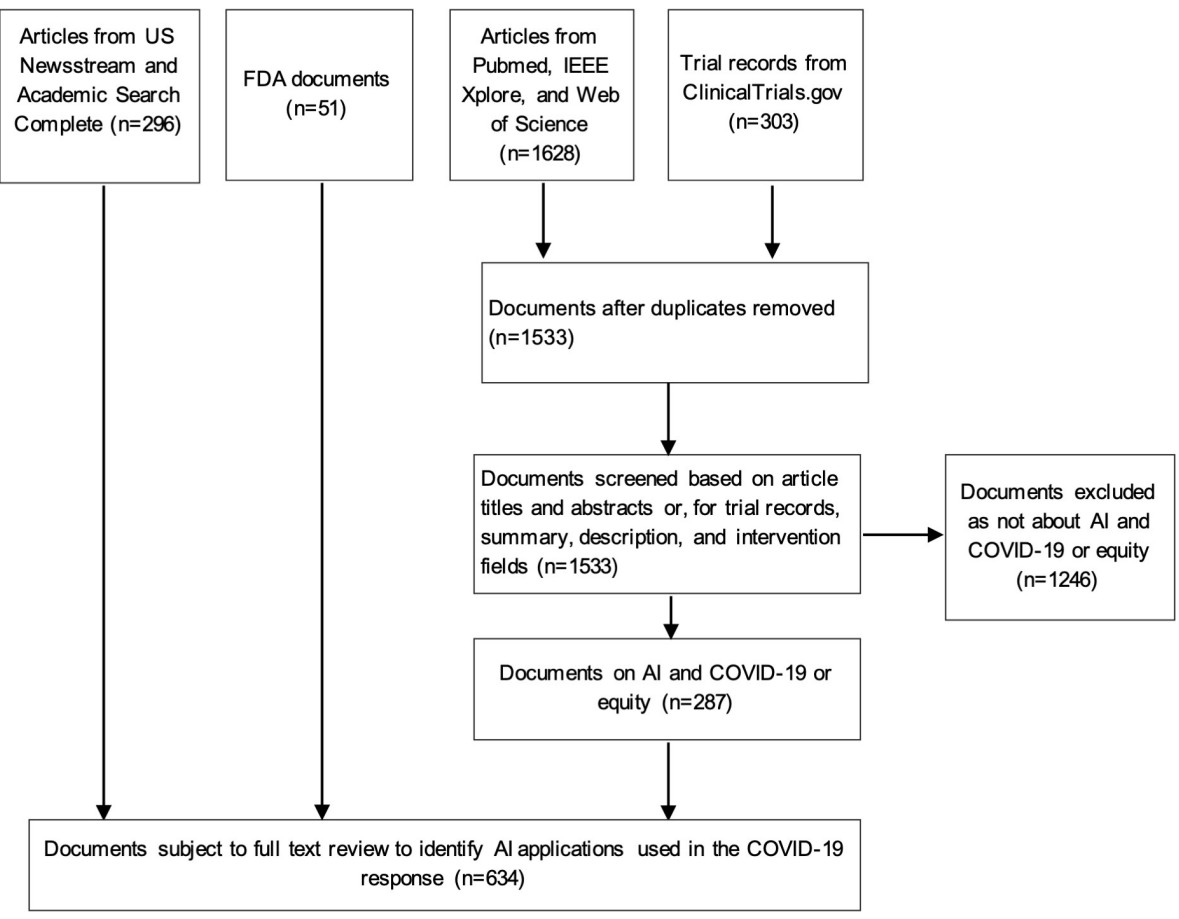

**Fig 1. PRISMA Flow Diagram.**

risk score (n = 4). All lung evaluation applications were used by clinicians in hospitals (n = 20), with some x-ray applications also used in mobile screening vans (n = 2), quarantine centers (n = 2), or border facilities and prisons (n = 1).

**Table 1. Summary of applications used in clinical response to COVID-19.**

| Functional category | Description |
| --- | --- |
| Lung evaluation | We identified 20 applications that assessed X-ray or CT images to evaluate the lungs of patients with suspected or confirmed COVID-19. |
| Symptom checker | We identified 12 patient-facing applications that prompted users to enter information on their symptoms and risk factors to provide a personalized COVID-19 infection risk assessment or care recommendations. |
| Patient deterioration | We identified 9 applications that monitored patients with COVID-19 for signs of clinical deterioration to inform care escalation decisions. |
| Infection likelihood | We identified 8 applications that predicted a patient's likelihood of COVID-19 infection to inform testing, triage, and diagnosis. |
| Disease severity | We identified 5 applications that predicted a patient's likelihood of experiencing severe disease to inform COVID-19 testing, triage, and treatment. |
| Other | We identified 12 applications that performed various other clinical functions. These include applications that assist in image acquisition, evaluate cardiac function, predict response to treatment, detect immune response, or assess breathing tube position. |

The proposed benefits of these applications included informing patient treatment (n = 18), informing diagnosis of COVID-19 (n = 16), speeding up patient evaluation (n = 14), assisting triage (n = 13), conserving staffing resources (n = 8), reducing disease transmission (n = 7), or replacing unavailable RT-PCR-based screening (n = 5).

## Symptom checkers

Symptom checkers analyzed patient-reported demographics, risk factors, and/or symptoms to provide a personalized COVID-19 risk assessment or care recommendations. Six symptom checkers interpreted user-entered free text inputs as part of this assessment. Symptom checkers were accessed via online website (n = 10), smartphone app (n = 4), mobile texting (n = 1), or voice assistant (n = 1). Most (n = 8) provided recommendations for when patients should seek further care, including 2 which offered to directly connect high-risk patients to a clinician for telehealth consultation. One symptom checker assessed the likelihood of post-acute sequelae of SARS-COV-2 infection (PASC), or "long COVID", in addition to active COVID-19 infection.

## Patient deterioration

Patient deterioration applications monitored COVID-19 patients for potential clinical deterioration to inform care escalation decisions. All sought to detect changes in patient vital signs. Three applications used data from wearable monitors. One application also predicted risk of respiratory failure and hemodynamic instability. One application estimated respiratory rate, heart rate, and movement from signals received by an under-mattress sensor. Another application estimated breathing patterns, movement, and sleep stages from an in-room wireless transmitter/receiver. Four applications used patient demographics and other electronic health record (EHR) data to inform their assessment.

Clinicians used these applications in hospitals (n = 6), assisted living facilities (n = 1), or to remotely monitor patients at home (n = 3). The proposed benefits of these applications included informing patient treatment (n = 9), improving patient safety (n = 9), reducing disease transmission (n = 7), and conserving staffing resources (n = 3).

## Infection likelihood

We found several different types of applications that assessed likelihood of COVID-19 infection.

Two applications predicted likelihood of COVID-19 infection based on volatile organic compounds present in an individual's breath. One was used by professional screening personnel in public settings and the other at border facilities. Their proposed benefits included informing COVID-19 diagnosis, speeding up patient evaluation, and lowering costs.

Two applications predicted RT-PCR results for individual test samples to optimize pooled testing. One application based its prediction on geographically aggregated data while the other used patient demographics and EHR data including free text clinician notes. Their proposed benefit was to conserve testing resources.

Two applications were used in clinical settings to triage patients for testing, one based on geographically aggregated data and the other on patient demographics, vital signs, and blood test results. Their proposed benefits included reducing disease transmission, speeding up patient evaluation, conserving staffing resources, and improving patient safety.

One application predicted the likelihood of COVID-19 infection by detecting a luminescent signal on a rapid antigen test strip. This application was designed for use by health care professionals and its proposed benefit was reducing errors in interpreting test results.

One application predicted the likelihood of COVID-19 infection using voice audio from calls to emergency services to assist in triage.

## Disease severity

Applications that predicted a patient's risk of severe COVID-19 were used to inform treatment (n = 3), prioritize patients for evaluation (n = 2), testing (n = 2), or vaccination (n = 1). All these applications based their predictions on patient demographics and medical records. One also used patient vital signs and blood test results. Four were used by clinicians in hospitals, outpatient clinics, or telehealth settings. One was used by professional call center personnel contacting patients by phone. Their proposed benefits included assisting triage (n = 3) and conserving testing resources (n = 2).

## Other

Other applications were used to perform a variety of functions in the COVID-19 response, including assisting image acquisition (n = 4), detecting immune response (n = 2), predicting response to treatment (n = 2), and performing other functions (n = 4).

Two image acquisition applications used cameras to detect anatomical landmarks and adjust patient position for CT scanning. One application evaluated image quality and predicted optimal ultrasound device manipulations to guide non-specialist clinicians conducting echocardiograms. One application generated Spanish-language audio instructions from previously translated text for patients undergoing chest x-rays.

Two applications detected signs of recent or prior COVID-19 infection based on detecting either antibodies or T-cells in patient blood samples, which could inform the assessment of PASC [27].

One application that predicted patient survival in response to treatment was used to prioritize COVID-19 patients to receive lung transplants. Another application was used to predict patient response to hydroxychloroquine.

One application analyzed chest x-rays to evaluate endotracheal breathing tube position. One application used echocardiography to estimate left ventricular ejection fraction for patients with potentially degraded cardiac function as a result of acute COVID-19 infection or PASC. One application identified barriers to hospital discharge using patient demographics and EHR records. One application provided a geographically aggregated measure of health disparities which were used to prioritize patients to receive a scarce medication, remdesvir.

## AI methods used in applications

Twenty-six applications used neural networks, mostly to interpret images for lung evaluation (n = 18), cardiac evaluation (n = 1), evaluation of breathing tube position (n = 1), or to assist in imaging acquisition (n = 2). Neural networks were also used to interpret wireless signals (n = 1), unstructured text (n = 2), or to generate voice audio from text (n = 1). Some neural network applications used image overlays to aid interpretation (n = 11), and one application generated a text explanation of its analysis using the GPT-3 natural language model [28].

Another six applications used advanced tree-based methods, specifically gradient-boosted trees (n = 5) and random forest models (n = 1), to analyze blood test results (n = 4), vital signs (n = 3), or patient demographics and electronic health records (n = 4). These predicted disease severity (n = 3), infection likelihood (n = 1), immune response (n = 1), or response to treatment (n = 1). Three of these applications used Shapley's Additive Explanations (SHAP) values to improve interpretability.

Seven applications used traditional supervised machine learning methods, including logistic regression (n = 4), unspecified regression (n = 2), Cox proportional hazards (n = 1), or linear discriminant analysis (n = 1). These applications based their predictions on patient demographics and EHR records (n = 5), blood test results (n = 2), vital signs (n = 2), or breath profiles (n = 1).

Two applications used unsupervised learning, including factor analysis and principal components analysis.

We did not find information on type of AI used for 25 applications, though these were described as using AI or ML. This included most symptom checkers (11 of 12), patient deterioration (6 of 9), and infection likelihood applications (5 of 8).

Three symptom checkers and one lung evaluation application were described as continuously learning from interpretation of new data. Three lung evaluation applications used AI-based natural language processing to automate labeling of training or validation datasets.

## Deployment date, location, and extent of use

We were able to determine the date (n = 64) and location (n = 62) of the documented deployment in the COVID-19 response for most applications (Fig 2). Most applications were deployed early in the pandemic, from January-March 2020 (n = 27) or April-June 2020 (n = 16). Fewer were first deployed from July-December 2020 (n = 10) or in 2021 (n = 11).

Twenty-five applications' first confirmed deployment was in the U.S., 12 in China, 19 in other high-income countries (HICs), and 6 in other low and middle-income countries (LMICs). All applications that first deployed in China did so from January-March 2020. Initial deployments in the U.S. and other countries were more spread out over time.

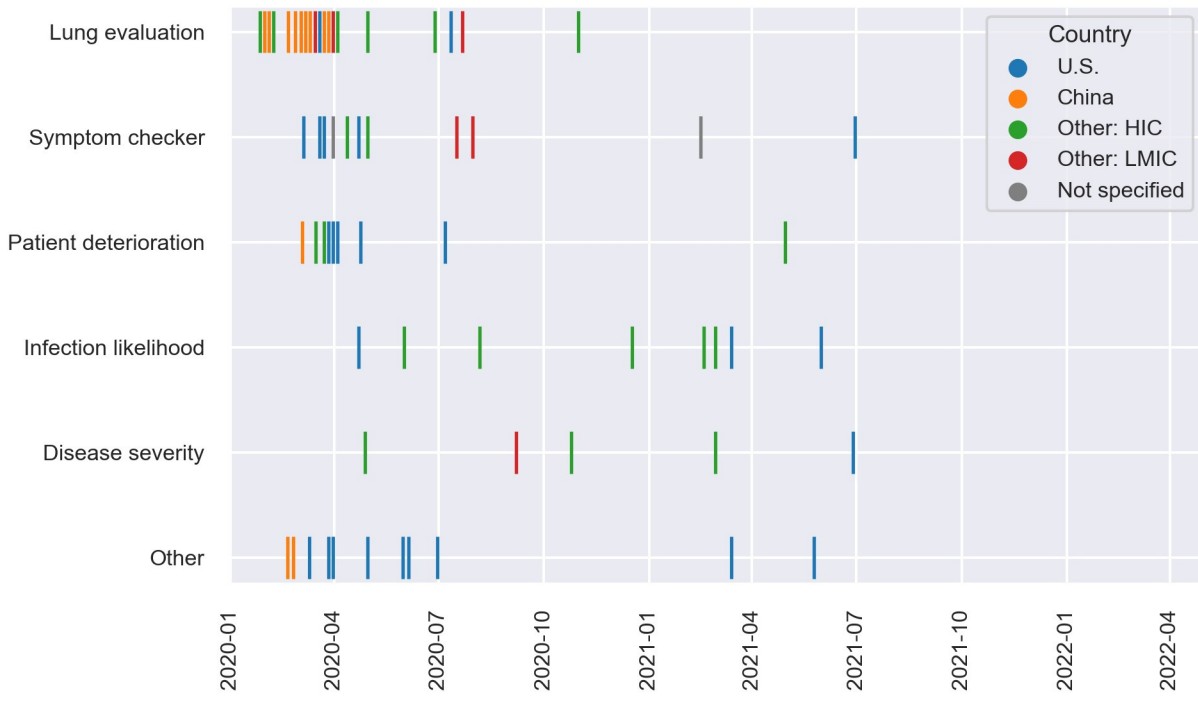

**Fig 2. Date and country of first confirmed deployment in the COVID-19 response, by application category.** Note: Dates have been jittered up to 12 days within the same month.

Most lung evaluation (13 of 20) and patient deterioration applications (5 of 9) were deployed from January-March 2020. Most symptom checkers were deployed between January-June 2020 (7 of 12). Half of infection likelihood applications were deployed in 2021 (4 out of 8). Most applications in the 'other' category were deployed between January-June 2020 (9 out of 12).

We found information on the scale of use for the majority of applications (n = 45) and country of use for most applications (n = 62) (Fig 3). Applications deployed in China had the highest usage, with half (6 of 12) used over 10,000 times and a quarter (3 of 12) used over 100,000 times. Lung evaluation applications tended to report higher usage than other application types. Patient deterioration applications tended to report lower usage or did not report usage at all.

Most applications were used in either the U.S. (n = 31) or China (n = 12), with applications also used in other HICs (n = 27) or LMICs (n = 7). Applications used in China were almost entirely for lung evaluation (9 of 12) as were those used in other LMICs (5 of 7). Applications in the U.S. included a large number focused on patient deterioration (8 of 31) and very few lung evaluations (2 of 31) and disease severity applications (1 of 31). The great majority of applications in the 'Other' category were used only in the U.S. (10 of 12), including all applications used to predict response to treatment, detect immune response, or assess breathing tube position. Most disease severity applications were used in other HICs (4 of 5).

## FDA review and application predecessors

As of December 2021, four applications had received FDA emergency use authorizations (EUAs), 11 applications had been cleared for use under the FDA 510(k) pathway for products that are "substantially equivalent" to already-approved devices [29], and one had been cleared as a novel device under the De Novo pathway. FDA-cleared or -authorized applications included most in the patient deterioration category (6 of 9), many in the 'other' category (6 of 12), and a small number of infection likelihood (1 of 8) and lung evaluation (2 of 20)

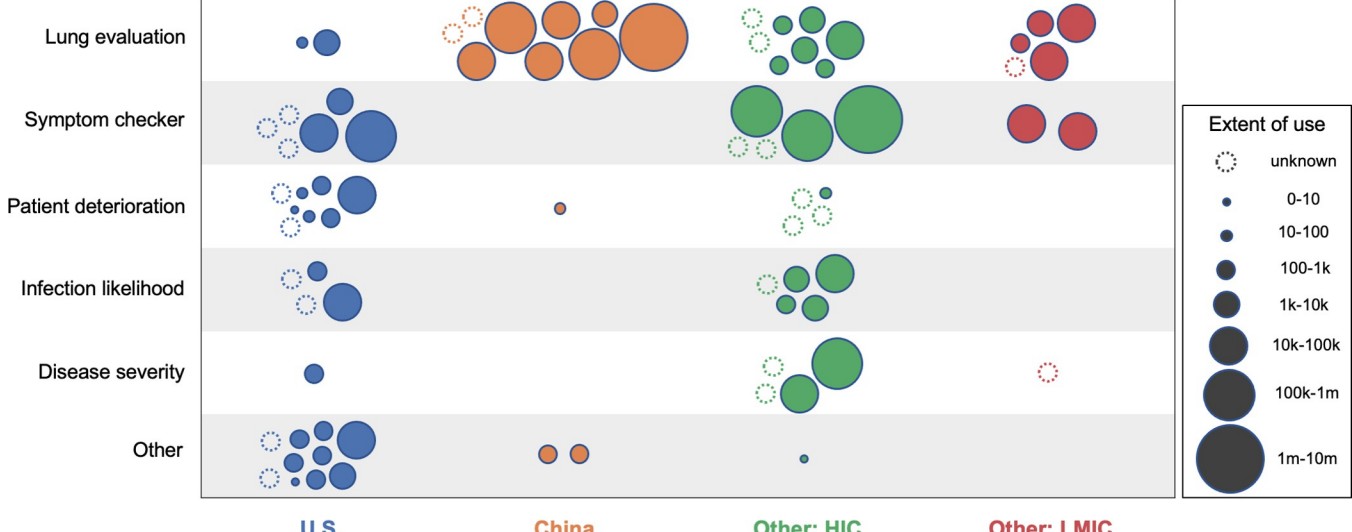

**Fig 3. Extent and location of use in the COVID-19 response, by application category.** Note: Scale of use reflects numbers of patients or related proxy measures. 10 applications were used in multiple country categories and are represented by multiple bubbles. We were unable to find exact country or scale of use for 3 online symptom checkers; these are represented by 'unknown' bubbles assigned to the country category where they were developed.

applications. The U.S. Center for Medicare and Medicaid Services designated 1 application eligible for product-specific Medicare reimbursement in 2021 [30].

Seventeen applications were already in use prior to the pandemic and were first deployed in the COVID-19 response without modification. These included most of the patient deterioration applications (7 of 9), a few lung evaluation applications (4 of 20), and several applications in the 'Other' category (6 of 14). Four of these applications were later modified to specifically address COVID-19, including three lung evaluation applications and one application in the 'Other' category. An additional 17 applications, including 5 lung evaluation and 10 symptom checkers, were adapted to address COVID-19 from applications used prior to the pandemic.

Of the remaining 32 new applications not directly related to a pre-pandemic product, at least three have additionally been used to address health conditions other than COVID-19, all in the U.S. These three saw their deployment accelerated due to the pandemic, including an application that assesses endotracheal breathing tube position deployed under blanket FDA guidance expanding use of applications without review, an application that assists in image acquisition deployed under an expedited 510(k) application, and a patient deterioration application rapidly deployed by a health system to address the pandemic.

## Evidence

We found that two applications whose use predated the pandemic were the subject of ten or more evaluation studies. For these we limited our review to the five studies that appeared most relevant to these applications' use in the COVID-19 response. We combined these 10 and all 71 evaluations found on the other AI applications for a total of 81 study documents included our review. We counted documents that evaluated two applications or that contained both validation and non-validation ('other') studies as two separate studies, resulting in a total of 90 distinct evaluation studies of AI applications.

These 90 studies provided evidence on 42 out of 66 applications (Fig 4). 39 applications received support from one or more evaluations. We found no publicly available evidence on the other 24 applications. This included most symptom checkers (9 out of 12) and patient deterioration applications (6 out of 9). 35 applications were examined in peer-reviewed articles and 7 applications were only evaluated in grey literature. 23 applications were the subject of a single study, and 19 applications were the subject of multiple studies.

Most studies provided evidence in support of application use (n = 73). Of the 23 studies that provided the sole evidence regarding an application's performance, 21 supported application use and 2 were opposed. Applications that were the subject of multiple studies were either supported by all (n = 12), received a mix of supportive and neutral evidence (n = 3), were supported by some studies and opposed by others (n = 3), or received a mix of neutral and opposed evidence (n = 1).

60 studies were written by authors affiliated with application developers; all but one supported application use. We found 30 independent evaluation studies, of which 27 were peer-reviewed. 14 of the independent evaluations supported application use, 7 were neutral, and 9 opposed. Only 10 applications received support from one or more independent peer-reviewed evaluation studies.

Most studies sought to validate the predictive performance of an application by measuring its agreement with a reference standard or its significance in predicting an outcome of interest (n = 73). 18 of these were prospective validation studies using data collected at the same time as the application was being tested. 44 studies were external validation studies that used a dataset independent from the data used to develop the application. 11 studies were internal validation studies that assessed application performance using the development dataset.

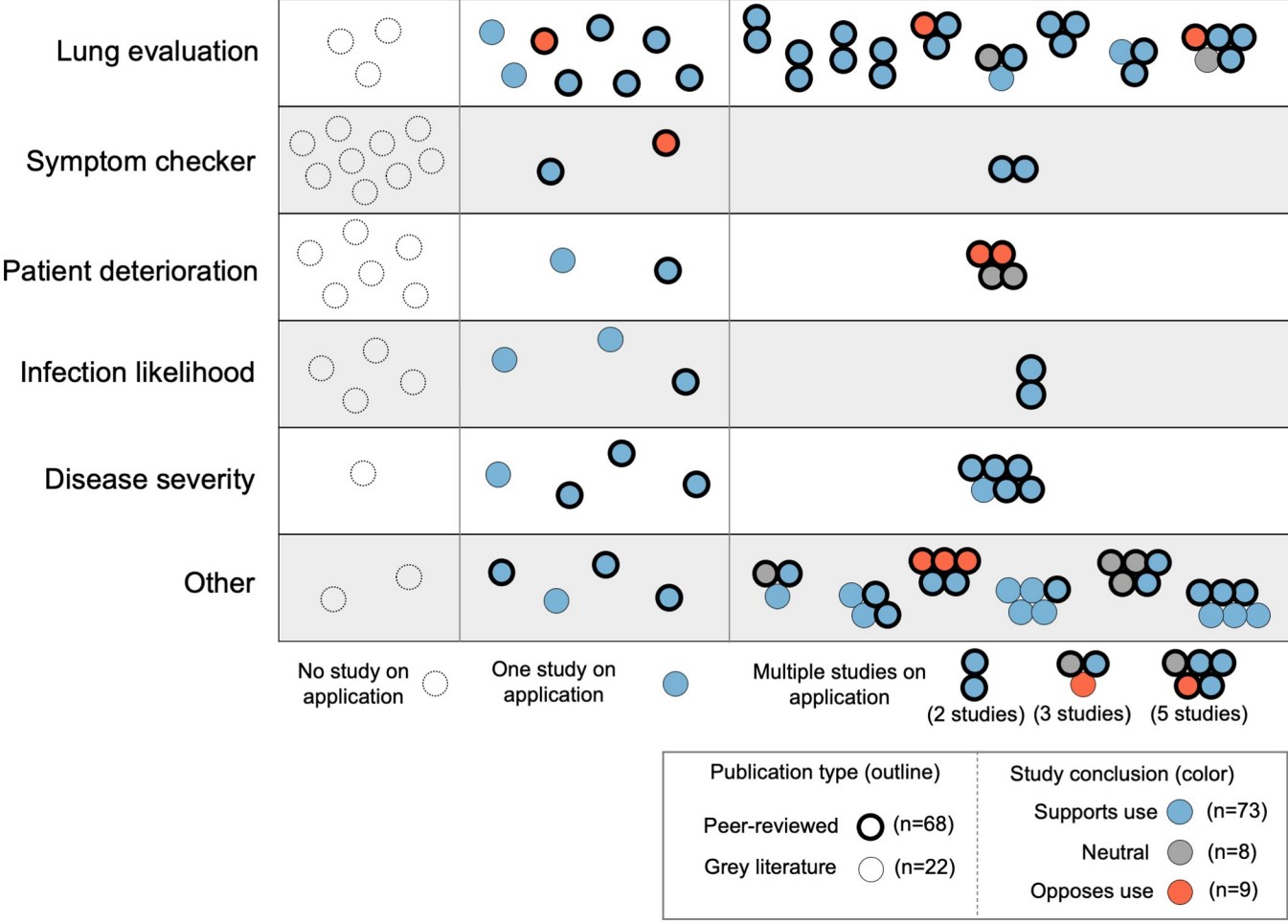

**Fig 4. Evaluation studies of applications, by study conclusion and publication type.** Note: Multiple studies on a single application are represented by a cluster of bubbles. Nine documents are each represented by two separate bubbles in this figure, including 3 documents that present evidence on two applications and 6 that presented evidence from both a validation study and a non-validation ('other') type of study.

17 studies did not validate an application's predictive performance and instead examined how an application's use affected clinical processes or outcomes. Most of these studies examined the quality of medical images obtained with application assistance (n = 7) or the effect of application use on time required for patient evaluation (n = 6). Two studies examined patient or clinician perceptions of application use. Another study examined the impact of AI-assisted CT scanning on patient radiation exposure. One study examined the impact of application use on cost of care.

Only a single study, which used a pre-post study design and measured mortality rates, measured the impact of an application's use on patient health outcomes. We found no randomized controlled trials that measured the impact of any application's use on health outcomes.

## Discussion

In this scoping review of academic and grey literature we identified and described a diverse set of 66 AI applications deployed in the clinical response to COVID-19. To our knowledge, no other study has performed such a comprehensive review or identified such a large number of clinical AI applications used to address the pandemic. The majority of the AI applications included in our analysis are not covered in previously published reviews [11,16,18].

## Some applications were used extensively

Our analysis supports a balanced view of AI applications' use in addressing COVID-19. Contrary to previously published statements that no AI model has been "widely adopted" [14] or "made a real difference" [31] in the pandemic response, we found that a few AI applications were used extensively. These included: several lung evaluation applications deployed early in China, symptom checkers used in the U.S. and other HICs, a disease severity model applied across the U.K.'s National Health System, and a predictive model used to inform all lung transplant allocation decisions in the U.S. and in several other countries.

At the same time, our analysis suggests that hopes for AI to significantly change the course of the pandemic have not been fulfilled [32,33]. A majority of AI applications we identified were used to an unknown or limited extent. We found no evidence of a major surge in AI adoption in clinical care due to the pandemic.

We also found little evidence to support the claim that "the pandemic has accelerated innovation of AI at a remarkable speed," [34] at least not in terms of AI being deployed in clinical care. The majority of applications used in the clinical response to COVID-19 either existed prior to the pandemic or were adapted from a pre-existing application. The pandemic did accelerate the deployment of a small number of new clinical AI applications that have uses beyond COVID-19, including applications to assess proper breathing tube placement, assist in image acquisition, and aid in echocardiography. Yet neither these nor applications developed solely to address COVID-19 represent developmental milestones for the underlying technologies they rely on. Though recent innovations in AI-based natural language processing [35], image generation [36], multi-purpose [37], and multi-trillion parameter models [38] have received extensive media attention, we found only a single application that used one of these new technologies, in this case the GPT-3 natural language model [39].

Our examination of where AI applications were used supports others' observations of a "digital divide" between higher and lower income countries in access to health technology [40,41]. Within the U.S. and other HICs, a wide variety of AI applications were used to assist the clinical response to COVID-19. Within China and other LMICs, however, we found very few deployed AI applications apart from those used in lung evaluation. This unequal global distribution of AI applications used in the clinical response to COVID-19 likely reflects broader disparities in access to health care.

## We found limited independent evaluations and no clinical trials on AI applications

A wide range of proposed benefits have been used to justify the deployment of AI applications used in the COVID-19 response. As a scoping review, we did not attempt to determine the extent to which these AI applications have delivered such benefits in practice, or resulted in any unintended harms, during the pandemic. We did, however, examine the availability of evaluation studies needed to make such a determination.

While a majority of applications were evaluated in peer-reviewed articles, several were only evaluated in grey literature such as FDA documents or academic pre-prints. Most evaluations provided evidence in support of application use, while a much smaller number opposed use.

However, only a small number of applications were independently evaluated in peer-reviewed studies, and these evaluations supported application use much less often than those written by authors affiliated with application developers. We found no evaluations for over a third of applications, including most applications within the symptom checker and patient deterioration categories. The absence of evaluations on the latter is particularly concerning,

given that such applications are often used to inform care escalation decisions for patients in critical condition.

Most evaluations focused on measuring an AI application's predictive validity, in other words, the accuracy or significance of the clinical information that the application provides. We found only a small number of prospective validation studies, consistent with others' findings that these more robust study types are not available for most clinical AI applications [1,11]. An even smaller number of evaluations examined the impact of using an application on clinical processes or outcomes. Surprisingly, only one study examined how an application's use affected patient health outcomes, and we found no randomized controlled trials on any application.

The lack of clinical trials examining AI applications in the COVID-19 response is striking, given the prominence of recent calls to carry out more such studies [42–44]. This may be because none of the AI applications we identified have a direct, independent impact on patient health that would be easy to measure and attribute to their use. Rather, these AI applications, similar to those used in clinical care beyond the pandemic response [45,46], are mostly used to inform diagnosis, prognosis, treatment, and triage decisions as part of a broader clinical management pathway.

Despite recent efforts to encourage clinical trials of AI applications and standardize their reporting [45,47], there is little guidance on how to conduct trials for the majority of applications whose impact on patient health comes via their influence on decisions made within a larger clinical pathway. Guidance on conducting clinical trials to examine the impact of diagnostic interventions [48,49], non-pharmacological treatments [50], or quality improvement interventions [51,52] might be adapted to inform evaluation of AI applications that perform similar functions. New guidance that addresses the unique ethical and practical challenges of conducting clinical trials of AI-based triage applications may be needed, given that their use determines which patients receive a scarce care intervention and which do not.

Finally, a principal challenge for AI developers is convincing clinicians that the application should be adopted. There are several reasons this challenge exists. For example, 1) the application may not offer a clear benefit above and beyond "usual care" if it is developed to approximate human decision-making, 2) clinicians may disregard the application's conclusion if the algorithm is opaque, and 3) adoption of the application may not be feasible in the clinical environment if it was not developed with clinical workflows in mind. Because of these factors, clinicians should be involved with AI application development through all stages, and research into the implementation of applications should be undertaken so that they can be modified to optimize adoption in clinical settings.

## Study limitations

Our study has several limitations. First, we combined systematic searches of academic and grey literature databases with more ad hoc methods including citation chain following, targeted Google searches, and eliciting document recommendations from stakeholder representatives. While these methods supported the discovery and review of a large and diverse set of academic and grey literature documents, they also hinder study reproducibility.

Second, we relied on publicly available documents to determine the extent of application use, and we only counted usage measures that corresponded to numbers of patients. This means that we underestimated the extent of AI deployment for applications without published information. Third, our review is undoubtedly missing some AI applications used in the clinical response to COVID-19, particularly those: (a) used outside the U.S. or in non-English speaking countries; (b) not explicitly labeled AI or ML but that nevertheless used traditional or

unsupervised AI methods; (c) whose use in clinical practice was not publicly documented; or (d) deployed later in the pandemic.

Fourth, we did not formally assess either performance metrics or risk of bias in the evaluation studies we found. Fifth, we limited our examination of regulatory status to applications reviewed by the U.S. FDA; a more comprehensive review of this subject would consider regulatory actions in China, the European Union, and other jurisdictions. Sixth, our analysis focuses solely on clinical applications. A fuller accounting of AI in the pandemic response would also consider applications in public health [11], vaccine development [53], drug repurposing [54], and viral protein mapping [55]. We are examining the use of AI in the public health response to the COVID-19 pandemic in a forthcoming report.

## Conclusions

We searched a wide range of sources to identify a larger number of AI applications used in the clinical response to the COVID-19 pandemic than reported previously. We report on these applications' location, time, and extent of deployment as well as their relation to pre-existing AI tools. We also report on their function, proposed benefits, type of AI, data inputs, and setting as well as the extent of available evidence supporting their use.

It is relatively easy to develop and test a new AI application and publish the results. It is much harder to deploy that application in clinical practice. The number of machine learning practitioners and journals, together with the availability of data, computing infrastructure, and open-source algorithms, has resulted in a proliferation of academic articles presenting new clinical AI models, including to address COVID-19. However, fewer of these applications have been used in clinical practice, and many of those that have been used are not evaluated in academic literature. Other reviews have solely focused on academic publications and do not differentiate between proposed models and those deployed in actual practice, and thus provide an incomplete picture of how emerging digital health technologies such as clinical AI are integrated into patient care.

AI is still an emerging technology in health care, with growing but modest rates of adoption in real-world clinical care. The COVID-19 pandemic showcased the range of applications to which it is put as well as the limited evidence available on most applications that have entered clinical use. The lack of adoption of AI may be due, in part, to the failure to study the barriers and facilitators to implementation of AI by clinicians. Future research should focus on implementation in health care settings and measure the impact of AI applications on patients' health outcomes. More independent evaluations are needed, particularly on understudied applications such as AI used to predict patient deterioration, that are likely to be adopted for use beyond the pandemic.

## Supporting information

**S1 Appendix. Stakeholder interview protocol and literature search documentation.**
(DOCX)

**S1 Table. AI applications used in the clinical response to COVID-19.**
(XLSX)

**S2 Table. Evaluation studies on AI applications used in the clinical response to COVID-19.**
(XLSX)

## Author Contributions

**Conceptualization:** Sean Mann, Carl T. Berdahl, Lawrence Baker, Federico Girosi.

**Data curation:** Sean Mann, Carl T. Berdahl, Lawrence Baker, Federico Girosi.

**Formal analysis:** Sean Mann, Carl T. Berdahl, Lawrence Baker, Federico Girosi.

**Funding acquisition:** Sean Mann, Federico Girosi.

**Methodology:** Sean Mann, Carl T. Berdahl, Lawrence Baker, Federico Girosi.

**Validation:** Carl T. Berdahl.

**Writing – original draft:** Sean Mann, Federico Girosi.

**Writing – review & editing:** Sean Mann, Carl T. Berdahl, Lawrence Baker, Federico Girosi.

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
