## [Decision Letter · Decision Letter 0]

15 Aug 2022

PDIG-D-22-00192

Artificial Intelligence Applications Used in the Clinical Response to COVID-19: A Scoping Review

PLOS Digital Health

Dear Dr. Mann,

Thank you for submitting your manuscript to PLOS Digital Health. After careful consideration, we feel that it has merit but does not fully meet PLOS Digital Health's publication criteria as it currently stands. Therefore, we invite you to submit a revised version of the manuscript that addresses the points raised during the review process.

Please submit your revised manuscript within 60 days Oct 14 2022 11:59PM. If you will need more time than this to complete your revisions, please reply to this message or contact the journal office at digitalhealth@plos.org. Please include the following items when submitting your revised manuscript:

We look forward to receiving your revised manuscript.

Kind regards,

Danilo Pani, Ph.D.

Academic Editor

PLOS Digital Health

Journal Requirements:

1. Please provide separate figure files in .tif or .eps format and remove any figures embedded in your manuscript file. Please also ensure that all files are under our size limit of 10MB.

For more information about how to convert your figure files please see our guidelines: https://journals.plos.org/digitalhealth/s/figures

2. We have noticed that you have a list of Supporting Information legends in your manuscript. However, there are no corresponding files uploaded to the submission (S1 Appendix). Please upload them as separate files with the item type 'Supporting Information'.

Additional Editor Comments (if provided):

The manuscript is interesting although there are several similar works in the field. The authors are kindly requested to provide a response, both in the revised manuscript and in a rebuttal letter, to the reviewers' criticisms, in particular those of the most critical reviewer.

Reviewers' comments:

Reviewer's Responses to Questions

**Comments to the Author**

1. Does this manuscript meet PLOS Digital Health’s publication criteria? Is the manuscript technically sound, and do the data support the conclusions? The manuscript must describe methodologically and ethically rigorous research with conclusions that are appropriately drawn based on the data presented.

Reviewer #1: Yes

Reviewer #2: Yes

Reviewer #3: Yes

2. Has the statistical analysis been performed appropriately and rigorously?

Reviewer #1: N/A

Reviewer #2: N/A

Reviewer #3: Yes

3. Have the authors made all data underlying the findings in their manuscript fully available (please refer to the Data Availability Statement at the start of the manuscript PDF file)?

Reviewer #1: Yes

Reviewer #2: Yes

Reviewer #3: Yes

4. Is the manuscript presented in an intelligible fashion and written in standard English?

Reviewer #1: Yes

Reviewer #2: Yes

Reviewer #3: Yes

5. Review Comments to the Author

Reviewer #1: Nice review article on AI applications developed in response to the COVID pandemic. My main concerns regard the novelty and take-home message of the article as a whole. Otherwise the article appears well-written, methodologically sound, and clear in the presentation of its results. My consideration goes more into the potential impact of a work like this (especially when its novelty is not entirely clear compared to existing similar work), especially in this phase of the COVID pandemic where priorities and surveillance activities are shifting their focus vs the initial phases of the pandemic (20-21) where most of the work reviewed in the article was performed.

Major points:

- A good number of reviews (correctly referenced in the article under evaluation) exist on this topic, e.g. Wynants et al. How is this review different? What’s its added value when a living review is already available on the same subject, especially in this phase of the pandemic? Please clarify since this is a determining factor for evaluation the original contribution of this article.

Also, apart from their deployment, number of users, and geographical coverage, I was expecting to find some sort of conclusion/discussion on the benefits (health, socio-economic, for policymaking) of the AI component of such applications. Does the well-known quote “100s of AI applications were developed for COVID… and none of them helped” still stand? Otherwise, what’s the take-home message from the review? I think this should be better elaborated and clarified in the review.

- Would be worth to clarify what do you mean by “employed in the clinical setting”. Does policy-making fall into the scope? The search terms look quite generic in this regard. Later note: I think I found a better definition in inclusion criterion 1, but for the full-text screening stage. I think this should be clarified earlier and, more importantly, evaluated earlier than the full-text analysis.

Minor/suggestions:

- I suggest to use the PRISMA reporting guideline (especially its adaptation for scoping reviews https://www.acpjournals.org/doi/10.7326/M18-0850) and associated diagram in support of the documentation efforts of the review (e.g. instead of figure 1, it would be good to adopt the PRISMA standard, which is not so different anyways)

- I find figure 3 too dense and at times hard to read. I feel there’s too many different dimensions crammed into one visualization. Number of studies, color, pattern.. I had to go through the legends 3 times to fully grasp all of it. Consider simplifying or redesigning

- Despite the review identified several applications outside the US/North American space (e.g. China and, I assume EU, in HICs) only the FDA approval was analyzed and not other equivalent quality assurance and certification procedures that are valid in places other than the US (e.g, EU medical device directive, GDPR, etc.)

Reviewer #2: The authors carried out an extensive search for applications of artificial intelligence in the clinical response to the COVID-19 pandemic and report their findings in the article. The authors describe their search methodology in enough detail to be clearly understood. However, since it involved interviews with a limited number, albeit a diverse set, of health care stakeholders, citation chain following, and Google searches, the results are not really reproducible in detail, a fact which the authors readily admit in the "Study Limitations" section, whose inclusion is to be commended. Another strength of the article is that the Introduction clearly states the four-fold objectives of the present study. They also clearly lay out the criteria for an application to be included in their analysis as well as the template for extraction of information from these applications. 

Their results are presented in an organized fashion: they grouped the AI applications into 5 main categories as well as a catchall bucket. It gets somewhat tedious to read through the exhaustive enumeration of how many applications fall into further sub-divisions within each category. Tables might have been used to more succinctly present this information. An adequate number of figures is presented, although they vary in quality and readability. For example, from Figure 2, it is easy to discern date, but the thin colored lines representing each application make it difficult to tease out country. Figure 4 (mislabeled as Figure 3) is very clear but could use some summary statistics to allow the main messages to jump out to the reader: e.g., x% are peer-reviewed, y% are supportive, etc.

The Discussion section is refreshing to read. It summarizes the main findings and provides evidence for or against published statements that are usually accepted uncritically by most readers. For example, their findings support the notion of a "digital divide" between low-income and high-income countries, but contradict statements that AI models have not yet been used extensively, whereas in fact, some have found extensive use in the COVID-19 pandemic. They also do not find support for the often-heard claim that the pandemic has accelerated AI innovation. 

They make some very important observations in this and the Conclusions section. More peer-reviewed, independent evaluations of AI applications are needed. Guidance is lacking but needed for how to conduct trials for AI applications that impact patient health and how to measure this impact. They surmise reasons for the challenges that AI applications face in gaining adoption: 1) how to show benefit beyond standard-of-care, 2) algorithm interpretability, and 3) lack of regard for clinical workflow. The authors rightly advise that clinicians should be involved in all stages of AI application development and that research should focus on implementation in real health care settings. A clear understanding of barriers and facilitators will accelerate the process of AI adoption by clinicians.

Reviewer #3: This paper provides a competently done scoping review of the use of AI for largely diagnostic purposes during the start of the COVID-19 pandemic. The authors collected and reviewed a fairly large and representative sample of work. Their conclusions are based on this review and seems balanced - they note the adoption of AI for diagnostic use (clinical response) and critical evaluate claims for its efficiency (e.g. also noting possible sources of bias). Their conclusion that there is a lack of clinical trails and that more research is needed is solid and in my view an accurate assessment and necessary call. 

The paper is well written, concise and will be a key input into future studies into the clinical use of AI and related data-driven and digital technologies in healthcare management.

6. PLOS authors have the option to publish the peer review history of their article (what does this mean?). If published, this will include your full peer review and any attached files.

**Do you want your identity to be public for this peer review?** For information about this choice, including consent withdrawal, please see our Privacy Policy.

Reviewer #1: No

Reviewer #2: Yes

Reviewer #3: No

---

## [Editor Report · Decision Letter 1]

20 Sep 2022

Artificial Intelligence Applications Used in the Clinical Response to COVID-19: A Scoping Review

PDIG-D-22-00192R1

Dear Mr. Mann,

We are pleased to inform you that your manuscript 'Artificial Intelligence Applications Used in the Clinical Response to COVID-19: A Scoping Review' has been provisionally accepted for publication in PLOS Digital Health.

Best regards,

Danilo Pani, Ph.D.

Academic Editor

PLOS Digital Health

I think the authors addressed some concers coming from the revision process. Maybe something more would have been better, but I think the work can be published in any case.